# Vision Language Models are In-Context Value Learners

**Yecheng Jason Ma**[†,1,2], **Joey Hejna**[1,3], **Ayzaan Wahid**[1], **Chuyuan Fu**[1], **Dhruv Shah**[1], **Jacky Liang**[1],
**Zhuo Xu**[1], **Sean Kirmani**[1], **Peng Xu**[1], **Danny Driess**[1], **Ted Xiao**[1], **Jonathan Tompson**[1],
**Osbert Bastani**[2], **Dinesh Jayaraman**[2], **Wenhao Yu**[1], **Tingnan Zhang**[1], **Dorsa Sadigh**[1], **Fei Xia**[1]
[1]Google DeepMind, [2]University of Pennsylvania, [3]Stanford University
Correspond to: jasonyma@seas.upenn.edu, xiafei@google.com
Website and Interactive Demo: http://generative-value-learning.github.io

## Abstract

Predicting temporal progress from visual trajectories is important for intelligent robots that can learn, adapt, and improve. However, learning such progress estimator, or temporal value function, across different tasks and domains requires both a large amount of diverse data and methods which can scale and generalize. To address these challenges, we present Generative Value Learning (GVL), a universal value function estimator that leverages the world knowledge embedded in vision-language models (VLMs) to predict task progress. Naively asking a VLM to predict values for a video sequence performs poorly due to the strong temporal correlation between successive frames. Instead, GVL poses value estimation as a temporal ordering problem over shuffled video frames; this seemingly more challenging task encourages VLMs to more fully exploit their underlying semantic and temporal grounding capabilities to differentiate frames based on their perceived task progress, consequently producing significantly better value predictions. Without any robot or task specific training, GVL can in-context zero-shot and few-shot predict effective values for more than 300 distinct real-world tasks across diverse robot platforms, including challenging bimanual manipulation tasks. Furthermore, we demonstrate that GVL permits flexible multi-modal in-context learning via examples from heterogeneous tasks and embodiments, such as human videos. The generality of GVL enables various downstream applications pertinent to visuomotor policy learning, including dataset filtering, success detection, and advantage-weighted regression – all without any model training or finetuning.

## 1 Introduction

Predicting temporal progress from visual trajectories is an important task for embodied agents that interact with the physical world. A robot capable of generalizable progress estimation can in principle discern desirable and undesirable behaviors to learn visuomotor skills in new environments. This is most often studied in reinforcement learning literature (Schaul et al., 2015), where progress estimation is equivalent to universal value learning under specific choices of reward function. However, universal value estimation comes with a number of key challenges: (1) broad *generalization* to new tasks and scenes, (2) the ability to *accurately estimate state* in partially observed environments, and (3) temporal *consistency* (i.e. satisfying the Bellman equation) over long horizons. Most existing methods trained on relatively small amounts of vision-only data (Chen et al., 2021; Ma et al., 2022; Ahn et al., 2022) lack the semantic, spatial, and temporal understanding needed to ground task progress in the space-time manifold of video, preventing generalization. Moreover, they often reason over single frames, inducing a high-degree of uncertainty in partially observed environments which in turn can effect the consistency of predictions for poorly estimated states. However, these challenges are not insurmountable: modern vision language models (VLMs) exhibit marked generalization and reasoning capabilities, potentially making them useful for value estimation.

Though not often considered as candidates for value estimation, VLMs excel at its aforementioned core challenges. First, state-of-the-art VLMs have exhibited strong spatial reasoning and temporal understanding capabilities across various vision tasks (Nag et al., 2022; Chen et al., 2024; Hong et al.,

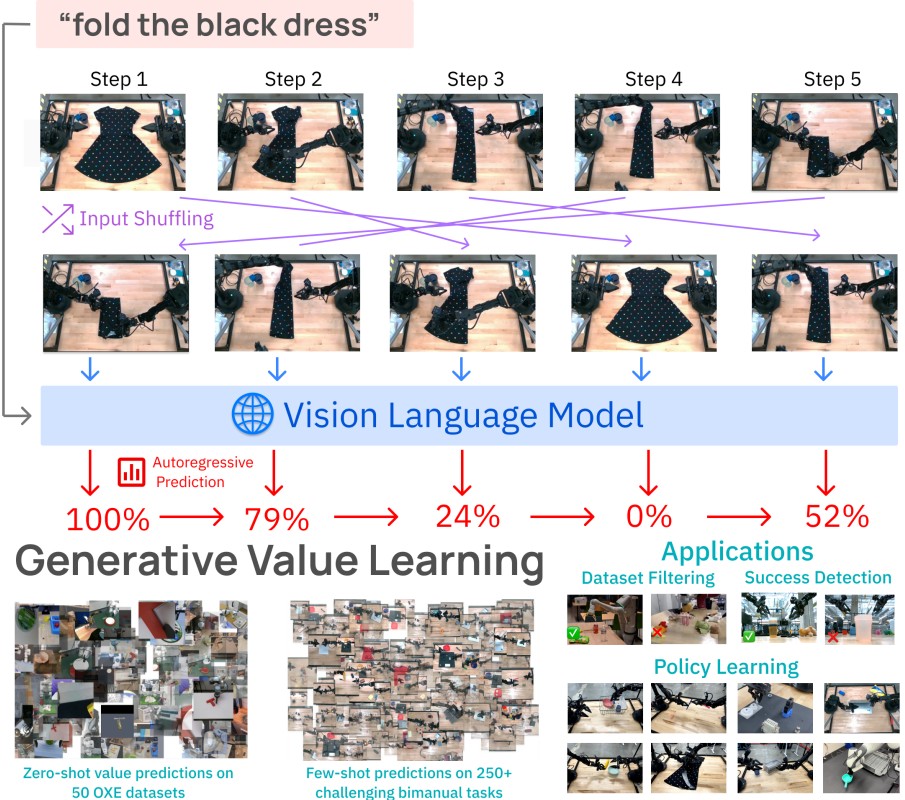

Figure 1: GVL auto-regressively predicts task completion percentage over shuffled frames, enabling impressive in-context value learning. GVL can effectively zero-shot and few-shot predict task progress on diverse and challenging real-world tasks; these capabilities enable expansive set of downstream applications, including dataset filtering, success detection, and policy learning.

2023; Gao et al., 2024), allowing them to *generalize* to novel scenarios. Second, large transformer-based VLMs have the requisite context window (GeminiTeam et al., 2024) to reason over large amounts of historical information to *accurately estimate state* from observation sequences when predicting task progress. Finally, VLMs make predictions *auto-regressively*, meaning they commit to their own outputs as inputs for subsequent predictions, imposing *consistency* constraints on long generations. For example, a VLM is unlikely to estimate that a task is 50% completed if it already has a 50% completion prediction in context. However, how exactly a VLM should be used to predict values is unclear. Empirically, we find that simply placing a video in-context and prompting the model to return progress predictions for each frame fails – our analysis suggests strong temporal correlations between successive frames often cause VLMs to produce uninformative monotonic values that disregard the actual quality of the trajectory and differences between frames (Section 4) – and a different approach is needed.

To effectively leverage the broad knowledge of VLMs, we introduce Generative Value Learning (GVL), a universal value estimation method enabled by long-context VLMs, which crucially operates over *shuffled* frames. At its core, GVL asks frozen state-of-the-art VLMs, such as `Gemini-1.5-Pro` (GeminiTeam et al., 2024), to auto-regressively predict the completion percentage of a task specified in natural language for a sequence of shuffled input video frames; see Fig. 1. Perhaps surprisingly we find that simply shuffling the frames of the input video effectively overcomes the strong implicit temporal bias found in video, enabling VLMs to generate meaningful values. While GVL is capable of generating values in a zero-shot manner, we find that the performance of GVL scales with examples via multi-modal in-context learning. Providing more examples of visual "unshuffling" in context increases performance, irrespective of the target embodiment. For example, human videos can improve GVL's performance on predicting robot task progress.

To facilitate large-scale value prediction evaluation, we additionally introduce a new evaluation metric, Value-Order Correlation (VOC), measuring how well predicted values correlate with the ground-truth timestep order in expert videos; as we will show, VOC is also a useful metric for

measuring dataset and trajectory quality, which allows GVL to be used for applications beyond value-based policy learning such as data quality estimation and success detection. We first evaluate GVL's value prediction quality with VOC on a large suite of real-world robotics datasets, spanning 51 datasets, 20 embodiments, and more than 300 tasks. This includes 50 datasets from Open-X (OXE) dataset (Padalkar et al., 2023) in addition to our own bimanual manipulation dataset containing 250 challenging real-world tasks on an ALOHA platform (Zhao et al., 2023), which are considerably longer horizon and more fine-grained than those in the OXE dataset. In aggregate, GVL exhibits strong zero-shot value prediction capabilities with highly positive VOC scores on most datasets; its performance further improves with various types of multi-modal in-context examples. Using GVL, we demonstrate scalable foundation model supervision for robot learning at various data abstraction levels. Specifically, GVL can help measure dataset quality in OXE. Second, it can be used for success detection, enabling imitation learning on mixed-quality datasets. Finally, the raw value estimates from GVL can be used for advantage-weighted regression for real-world offline reinforcement learning (Peters & Schaal, 2007; Peng et al., 2019).

In summary, our contributions are

1. Generative Value Learning (GVL), a universal value prediction framework via VLM in-context autoregressive value estimation on shuffled video frames.
2. An extensive evaluation on real-world datasets demonstrating GVL's zero-shot scalability and multi-modal in-context learning capabilities.
3. Demonstration that GVL can be used in downstream applications including dataset quality estimation, success detection, and advantage-weighted regression for real-world control.

## 2 RELATED WORK

**Reward and value foundation models.** Several works have tried to learn transferable reward and value functions from diverse data. Early works learned models using robot (Sermanet et al., 2016) or even human videos with discriminators (Chen et al., 2021), contrastive learning (Baumli et al., 2023) or offline RL (Ma et al., 2022; 2023a; Bhateja et al., 2023) to guide manipulation tasks. With the advent of recent language and vision foundation models, several works have integrated them into various robotic applications such as semantic planning (Ahn et al., 2022; Huang et al., 2023b; Singh et al., 2023; Zhang et al., 2023; Ding et al., 2023), imitation learning (Brohan et al., 2023; Szot et al., 2023), and symbolic programming (Tang et al., 2023; Liang et al., 2023; Singh et al., 2023; Wang et al., 2023; Huang et al., 2023a; Liu et al., 2023; Silver et al., 2023; Ding et al., 2023; Lin et al., 2023; Xie et al., 2023b). Most related to our work, LLMs and VLMs have been used as reward models. Kwon et al. (2023b); Mahmoudieh et al. (2022) use language models to provide reward values for RL agents, while Klissarov et al. (2023); Wang et al. (2024); Kwon et al. (2023b) use them to provide preference feedback. Ma et al. (2023b); Yu et al. (2023); Xie et al. (2023a) even have LLMs generate their code. These works use only the language capabilities of foundation models. More recent works directly use VLMs as zero-shot reward models (Rocamonde et al., 2023) or success detectors (Du et al., 2023; Guan et al., 2024). Critically, in these works the VLM acts only as an (often sparse) reward function which predicts success, and not a *value* function that predicts task progress. Though some works use chain-of-thought prompting (Venuto et al., 2024) or active learning (Kwon et al., 2023a), they generally do not make use of the autoregressive, long-context, or in-context learning capabilities of state-of-art VLMs. As a consequence, they often evalaute *reward* prediction only on simple and simulated tasks. To our knowledge, we are the first to demonstrate that VLMs are capable of generalizable per-frame value estimation on real world tasks which can be used for downstream tasks like dataset selection.

**In-context learning for robotics.** In-context learning has been explored in the robot learning literature, primarily focusing on action generation (Duan et al., 2017; Finn et al., 2017; Dasari & Gupta, 2021; Xu et al., 2022; Di Palo & Johns, 2024; Liang et al., 2024; Fu et al., 2024). However, all these prior works require explicit, and often extensive training, on their robot tasks in order to realize in-context learning capabilities, and generalization is achieved only on narrow distribution of tasks. In contrast, we demonstrate that visual value estimation already enjoys flexible multi-modal in-context learning from pre-trained VLMs without any robot specific fine-tuning.

## 3   GENERATIVE VALUE LEARNING

In this section, we introduce Generative Value Learning, GVL. At a high level, GVL frames value estimation as an autoregressive next-token prediction problem in which a VLM is tasked with outputting the task progress for a *batch of shuffled trajectory frames.*

**Problem setup.** We model robotics tasks as goal-conditioned partially observed Markov decision processes (Puterman, 2014): $\mathcal{M}(\phi) := (O, A, R, P, T, \mu, G)$ with observation space $O$, action space $A$, reward function $R$, transition function $P$, task horizon $T$, initial state distribution $\mu(o)$, and goal space $G$ that specifies the task semantically. Conditioned on a task $g$ an agent $\pi : O \to A$ aims to maximizes its value function, or the expected cumulative reward over the task horizon, $V^\pi(o_1; g) = \mathbb{E}_{\mu,\pi,P}[r(o_1; g) + \cdots + r(o_T; g)]$. However, reward and value functions can be difficult to define for robotics applications given their heterogeneity. Given this, a popular universal notion of value is task progress (Sermanet et al., 2016; 2018; Eysenbach et al., 2020; Tian et al., 2020; Lee et al., 2021). This kind of temporal value function maps an observation and goal specification to a real number between 0 and 1: $V : \mathcal{O} \times \mathcal{G} \to [0, 1]$, where initial observations of the environment have value 0 and goal-satisfying observations have value 1. Under this definition, an expert trajectory $\tau = (o_1, \ldots, o_T) \sim \pi_E$, has value function $V^{\pi_E}(o_t; g) = \frac{t}{T}$. In this work, our goal is to obtain such a temporal value function $V$ that can predict such task progress $v_1, \ldots v_T$ for each frame of video $o_1, \ldots, o_T$.

Though we seek to leverage priors imbued in large foundation models, as shown in Section 4 simply prompting a VLM with video frames fails to produce meaningful estimates. To make VLMs amenable to value prediction, we propose three key components that comprise the GVL method: 1) autoregressive value prediction, 2) input observation shuffling, and 3) in-context value learning.

**1. Autoregressive value prediction.** Traditionally, value functions $V(\cdot) : \mathcal{O} \to \mathbb{R}$ are trained to be self-consistent by enforcing the bellman equation

$$V^\pi(o_t) = R(o_t) + \mathbb{E}_{\pi, P}\left[V(o_{t+1})\right]. \tag{1}$$

When parameterizing a value function as a feed-forward neural network, this is typically done by minimizing the mean-squared error of the equality above. As values for different observations within the same trajectory are related via the bellman equation, the resulting value function remains consistent even if we query it with only a single observation. VLMs on the other hand are not inherently trained with any consistency objective. Thus, if we independently query a VLM with different observations from the same trajectory it is likely to produce inconsistent values. Our insight is that providing the entire trajectory as input instead of just a single observation offers VLMs greater opportunity to generate self-consistent value estimates. Concretely, given a language description of the task $l_{\text{task}}$ we ask the VLM to auto-regressively generate values given the entire video as context:

$$v_t = \text{VLM}(o_1, \ldots, o_T; v_1, \ldots, v_{t-1}; l_{\text{task}}), \forall t \in [2, T]. \tag{2}$$

We abbreviate this auto-regressive prediction process as $v_1, \ldots, v_T = \text{VLM}(o_1, \ldots, o_T; l_{\text{task}})$. This simple mechanism allows the VLM to attend to all previous predictions and frames when making the next value prediction, enabling it to produce globally consistent estimates over long-horizon sequences without needing to be trained like classical feed-forward value functions. Though this design choice enables VLMs to produce consistent values, it doesn't necessitate that the values are meaningful. Naïvely prompting a VLM in this manner tends to produce linear, monotonic value functions for every single video, regardless of optimality.

**2. Input observation shuffling.** Empirically we find that when presented a choronological sequence of frames VLMs discover the short-cut solution of outputting monotonically increasing values, often ignoring the task description or the actual quality of the trajectory. One hypothesis is that as VLMs are trained on ordered video frames for captioning and question answering, the chronology itself is a cue for downstream tasks unrelated to value prediction. As a consequence, model naïve prompting results in unfaithful low-quality value predictions. To break this temporal bias, we propose randomly shuffling the input frames. In this manner, GVL forces the VLM to pay attention to each individual frame and output faithful value predictions using all information provided in context. Concretely, GVL prompts a VLM as:

$$v_{\tilde{1}}, \ldots, v_{\tilde{T}} = \text{VLM}(o_{\tilde{1}}, \ldots, o_{\tilde{T}}; l_{\text{task}}, o_1), \quad \text{where} \quad (\tilde{1}, \ldots, \tilde{T}) = \texttt{permute}(1, \ldots, T). \tag{3}$$

where the permute operator randomly shuffles the temporal indicies. Note however, that we cannot shuffle *every* frame. If we do so, then the arrow of time in the original video can be ambiguous –

i.e., in many cases, the reverse video is also physically plausible, making is the ground-truth order impossible to predict. Thus, as in the above equation we condition the VLM on the first input frame allow it to use the first observation as an anchor point for all other shuffled frames.

**3. In-context value learning.** While auto-regressive prediction and shuffling are enough to obtain good performance, GVL can perform even better by leveraging the appealing properties of VLMs. Notably, large models often exhibit in-context learning, where tasks can be learned by simply providing examples (Brown, 2020). This enables flexible and versatile in context value learning, by which GVL's predictions can steadily improve by providing examples at test time without any model fine-tuning. In particular, we can simply prepend shuffled videos and their ground-truth task progress as in-context examples to boost the value prediction quality via few-shot learning:

$$v_{\tilde{1}}, \ldots, v_{\tilde{T}} = \text{VLM} \left( o_{\tilde{1}}, \ldots, o_{\tilde{T}}, l_{\text{task}} \mid \texttt{permute} \left( (o_1, v_1), (o_2, v_2), \ldots, (o_M, v_M) \right) \right) \quad (4)$$

As we show in Section 4, GVL benefits from flexible forms of in-context examples, including videos from unrelated tasks and even humans. Though GVL zero-shot is already effective across a broad range of tasks and robots, in-context learning can still realize substantial improvement on the most difficult bimanual dexterous tasks.

## 4    EXPERIMENTS

We conduct large scale experiments assessing GVL's value prediction generalization and in-context learning capabilities. Specifically, we study the following questions:

1. Can GVL produce zero-shot value predictions for a broad range of tasks and embodiments?
2. Can GVL improve from in-context learning?
3. Can GVL be used for other downstream robot learning applications?

In all our experiments, we use `Gemini-1.5-Pro` (GeminiTeam et al., 2024) as the backbone VLM for GVL; we ablate this model choice and find GVL effective with other VLMs as well. After thorough study of GVL's value prediction capabilities, we study several downstream applications in visuomotor policy learning, aiming to improve data quality at dataset, trajectory, and individual transition levels.

**Evaluation metric.** Our goal is to evaluate GVL value estimation at scale on as many robot datasets as possible, holistically testing its generalization capabilities and understanding its limitations. This makes it difficult to use traditional evaluation metrics for value functions, such as observing downstream learned policy performance, as they require value functions that are specifically trained or finetuned for individual tasks and embodiments. This quickly becomes very expensive for universal value functions that are intended for use across a large set of diverse real-world tasks and robots, many of which the practitioner may not have access to. Prior works on large-scale value learning have resorted to visually observing the smoothness of the value curve on expert trajectories as a qualitative "eye-test" for model generalization (Ma et al., 2022; 2023a; Karamcheti et al., 2023), but such evaluation is conducted on only few selected videos. We formalize and scale up this intuitive approach and introduce a lightweight, yet predictive method for evaluating value models: Value-Order Correlation (VOC). This metric computes the *rank correlation* between the predicted values and the chronological order of the input expert video:

$$\text{VOC} = \texttt{rank-correlation} \left( \texttt{argsort}(v_{\tilde{1}}, \ldots, v_{\tilde{T}}); \texttt{arange}(T) \right); \quad (5)$$

VOC ranges from $-1$ to $1$, where $1$ indicates that the two orderings are perfectly aligned. Expert quality demonstrations, by construction, have values that monotonically increase with time, and thus a good value model should have high VOC scores when evaluated on expert videos. On the other hand, fixing a good value model, low-quality trajectories should have low VOC scores. This is because sub-optimal trajectories often contain high repetition of visually similar frames due to the presence of redundant, re-attempt actions or poorly-placed cameras. As such, the values along the trajectories should not be monotonic, resulting in low correlation with the ground-truth timestep order. As we will show in our experiments, this value rank correlation metric has strong predictive power for the quality of the values as well as downstream policy learning performance, validating its usefulness as a standalone evaluation metric for value predictions.

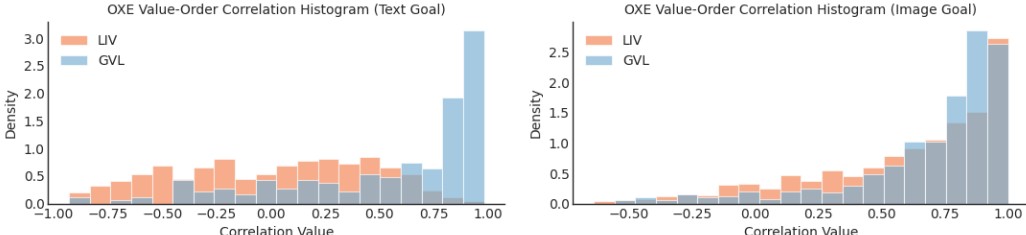

Figure 2: **Zero-shot value predictions on OXE datasets.** Left: GVL significantly outperforms LIV on datasets with language goals. Right: GVL still outperforms LIV on datasets with image goals despite solving the more difficult task of frame re-shuffling.

## 4.1 LARGE-SCALE REAL-WORLD EVALUATION

To study GVL's zero-shot value prediction capability, we evaluate its VOC on two large expert robotics datasets.

**Open X-Embodiment dataset.** First, we consider the Open X-Embodiment (OXE) dataset (Padalkar et al., 2023). an aggregation of trajectory data from 50 standalone academic robot datasets that consists of diverse tasks, robots, and camera viewpoints. For each of the 50 datasets, we randomly sample 20 trajectories and evaluate GVL zero-shot on each of the sampled trajectories. Note that not all OXE datasets have language task annotations, so we use the last frame of the trajectory as the goal specification when text annotation is not provided. To better contextualize GVL's value prediction quality, we compare to a state-of-the-art multi-modal value model **LIV** (Ma et al., 2023a), a contrastive vision-language model (Radford et al., 2021) fine-tuned with value learning objective on human videos for in-the-wild value estimation. LIV predicts the temporal value of an input observation by computing its embedding distance to the embedding of the goal image or task description.

For evaluation, we plot the histogram of all 1000 (50×20) Value Order Correlation (VOC) scores in Fig. 2, split by goal modalities. Given that most OXE datasets contain human-collected expert demonstrations, good value models should have high VOC scores; however, we acknowledge that there are sub-optimal trajectories within OXE that can introduce noise in our results; in Fig. 10 in Appendix F, we further evaluate the two methods on a subset of OXE datasets that is delegated to be high-quality to corroborate Fig. 2. After we first establish GVL as an effective universal value model. we will present how GVL can be used to detect low-quality data in Section 4.3. As shown in Fig. 2, on both goal modalities, GVL consistently generates VOC scores that heavily skew to the right, indicating that it is able to zero-shot recover the temporal structure hidden in the shuffled demonstration videos, i.e., coherent value predictions. GVL's performance is also markedly better than LIV on language goals (Fig. 2 left). Here, LIV's predictions are random, suggesting that its embedding space does not contain sufficient knowledge for predicting dense values for arbitrary unseen robot videos. On image goals, LIV's prediction problem is arguably simpler because an embedding space that simply captures image similarity can result in ascending values that correlate with timesteps. Even then, GVL generates better quality value predictions as judged by slightly higher VOCs (Fig. 2 right). In summary, GVL can indeed effectively utilize the world knowledge afforded by the backbone VLM to achieve effective value predictions zero-shot for the breadth of real-world robotic tasks and datasets.

**Challenging bimanual datasets.** OXE datasets primarily focus on simpler, short horizon single-arm tasks. To further stress test GVL, we evaluate on a new diverse dataset of 250 distinct household tabletop tasks on the bi-manual ALOHA systems (Zhao et al., 2023; Team et al., 2024a). This dataset includes highly challenging, long-horizon skills, such as removing three gears sequentially from a NIST board, folding a dress in eighth-fold, hanging a t-shirt on a cloth rack. See the bottom right of Fig. 1 for representative ALOHA tasks. For each task, we evaluate on 2 human teleoperated demonstrations to evaluate GVL and LIV zero-shot. The aggregate histogram over all 500 (250 × 2) VOC scores is illustrated in Fig. 3. As shown, GVL is capable of generating value predictions that are positively correlated on more than 60% of them with median VOCs of 0.12. This is promising, but worse than the performance on the OXE datasets. GVL and LIV exhibit similar qualitative trends, indicating that both methods now struggle with the complexity of this dataset. In the next section, we extensively explore how to improve GVL on this dataset using in-context learning techniques.

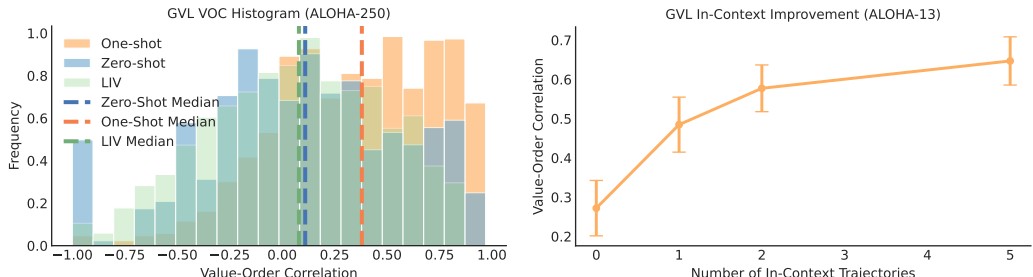

Figure 3: GVL scales up to 250 ALOHA bi-manual tasks and can improve with in-context examples.

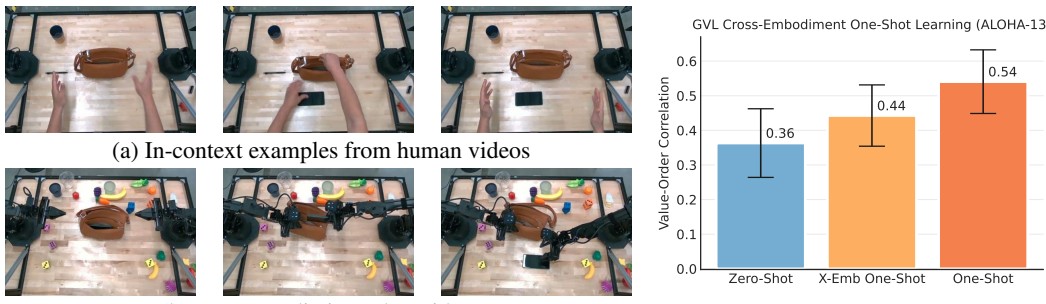

(a) In-context examples from human videos

(b) Target prediction robot videos

Figure 4: GVL benefits from cross-embodiment in-context learning capability: its value predictions can be improved by examples from human videos.

## 4.2 MULTI-MODAL IN-CONTEXT VALUE LEARNING

As the diverse ALOHA dataset is significantly more challenging, we explore whether GVL can benefit from in-context learning, where additional shuffled observation-value pairs are presented in the VLM context window (Eq. 4).

**Few-shot in-context learning.** First, we collect an additional demonstration for each of the 250 tasks and use its shuffled value-observation pairs as context for one-shot GVL value prediction for the same set of 500 evaluations. As seen in Fig. 3, with one in-context trajectory, GVL's performance substantially improves with 90% positive VOCs and a median VOC of 0.37. We further investigate whether performance can improve with more in-context examples on a represented subset of 13 tasks for which have more than 500 demonstrations; we refer to this subset as ALOHA-13. For these tasks, we evaluate few-shot GVL on 500 distinct trajectories per task with up to 5 in-context examples. The average VOCs over tasks and trajectories is shown in Fig. 3 (Right). We see that GVL demonstrates appealing in-context scaling as the average score steadily improves as we increase the number of in-context examples. Even with 5 in-context trajectories, meaning 150 total shuffled images, GVL is able to utilize its full context and exhibit strong generalization. This result demonstrates how state-of-art long-context-window VLMs, such as `Gemini-1.5-Pro`, can be re-purposed to make for general-purpose value functions with impressive test-time improvement capability, quickly mastering value predictions with minimal supervision.

**Cross-embodiment in-context learning.** Examples in-context are not limited to robot demonstrations. One advantage of GVL is that it can still benefit from in-context learning even when the demonstrations come from a different embodiment. Specifically, we record humans performing the same tasks as the ALOHA robot demonstrations and then use these human demonstrations as in-context examples for value prediction. As shown in Fig. 4, GVL with one cross-embodiment in-context example can effectively improve over its zero-shot counterpart. In the Appendix, we also show that GVL can similarly benefit from *cross-task* in-context learning. In conclusion, GVL presents a versatile framework for in-context value learning that can scale up to even the most challenging manipulation tasks.

## 4.3 GVL APPLICATIONS

As GVL can generate high-quality value estimates, it can be applied to a number of downstream tasks including dataset quality estimation, success detection, and weighted imitation learning.

**Dataset Quality Estimation.** Robotic action models are increasingly trained on large mixtures of datasets (Padalkar et al., 2023; Team et al., 2024b; Kim et al., 2024) and selecting the right mixture is critical for policy performance (Hejna et al., 2024). However, dataset mixing is often done in an ad-hoc fashion by visual inspection (Team et al., 2024b). Having validated that GVL is an effective zero-shot value model, we investigate whether we can in turn use GVL's VOC scores to determine dataset quality within OXE. To this end, for each OXE dataset in Fig. 2, we compute the average correlation scores for its sampled trajectories and present the ranking of the average score in Appendix C. In Table 1, we present a subset of selected representative large-scale datasets in OXE. We see that datasets have large spread in their VOC scores, but these scores are interpretable and match human intuitions. Specifically, datasets collected from human teleoperators with relative fixed camera placements, such as RT-1 (Brohan et al., 2022), Dobb-E (Shafiullah et al., 2023), and Bridge (Ebert et al., 2021; Walke et al., 2023), have high VOC scores, despite their diversity in scenes and tasks. In contrast, datasets with autonomous data collection via scripted motions or motor babbling, such as QT-OPT (Kalashnikov et al., 2018) and RoboNet (Dasari et al., 2019), contain high number of suboptimal trajectories that do not exhibit smooth temporal structure to be re-shuffled.

Interestingly, DROID (Khazatsky et al., 2024), a recent large household manipulation dataset is ranked very low, consistent with prior works (Kim et al., 2024) that found that removing DROID from large action model training improved final performance. After inspecting trajectories from DROID with a low VOC score from GVL we found that many have poor camera angles that do not capture robot motion or have the arm or manipulated objects heavily occluded.

| Dataset | Avg. VOC |
|---|---|
| RT-1 (Brohan et al., 2022) | 0.74 |
| Dobb-E (Shafiullah et al., 2023) | 0.53 |
| Bridge (Walke et al., 2023) | 0.51 |
| QT-OPT (Kalashnikov et al., 2018) | 0.19 |
| DROID (Khazatsky et al., 2024) | -0.01 |
| RoboNet (Dasari et al., 2019) | -0.85 |

Table 1: Average GVL VOCs on selected OXE datasets.

These observations indicate that GVL VOC can be indicative of dataset quality. In Appendix G, we show that GVL dataset quality estimation can effectively be used to generate effective co-training datasets from a raw, mixed-quality dataset such as DROID for real-world imitation learning.

**Success detection and filtered imitation learning.** Next we consider more granular intra-dataset quality control by investigating how GVL can be used as a success detector for trajectory filtering, enabling filtered imitation learning on mixed quality datasets. As discussed, good value models should return low

| Method | Accuracy | Precision | Recall |
|---|---|---|---|
| GVL-SD (Zero-Shot) | 0.71 | 0.71 | 0.71 |
| GVL-SD (One-Shot) | **0.75** | **0.85** | 0.70 |
| SuccessVQA (Du et al., 2023) | 0.62 | 0.33 | 0.73 |
| SuccessVQA-CoT | 0.63 | 0.44 | 0.68 |

Table 2: Comparison of VLM success detectors.

VOC scores on unsuccessful trajectories; in particular, it is difficult for GVL to re-shuffle frames within sub-optimal trajectories which often contain irregular or repetitive behavior . Thus, we can use GVL for success detection by filtering trajectories that have VOC scores below certain numerical threshold; we refer to this procedure as GVL-SD. We evaluate GVL-SD on six simulated bimanual dexterous manipulation tasks on the ALOHA system (see Fig. 7). Simulation is well-suited for this experiment because we can naturally control for data quality and reproducibility. More specifically, for each task, we construct a mixed quality dataset by rolling out a pre-trained policy of roughly 50% success rate for 1000 episodes, mirroring real-world autonomous data collection settings with high failure rate (Kalashnikov et al., 2018). We compare to **SuccessVQA** (Du et al., 2023), which poses success detection as a Visual-Question Answering problem. To ensure that the same amount of information is provided, we feed the full video sequence to the VLM; therefore, this baseline tests whether the VLM is equipped with video understanding capability good enough for out-of-the-box success detection. In addition, we consider **SuccessVQA-CoT**, which uses chain-of-thought prompting (Wei et al., 2022) to encourage the VLM to output intermediate textual reasoning outputs before providing the final success answer. Unless otherwise stated we use a VOC threshold of $0.5$ for GVL-SD.

For all methods, we report the accuracy, precision, and recall in Table 2. GVL-SD consistently outperforms or matches SuccessVQA on all classification metrics. In particular, SuccessVQA has

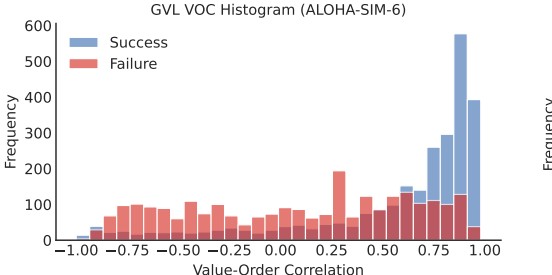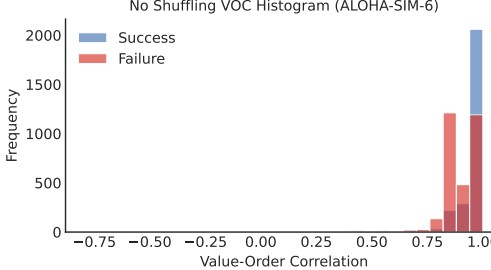

Figure 5: **GVL for Success Detection.** Left: GVL behaves qualitatively differently on successful and failed trajectories. Right: GVL (No-Shuffling) loses discriminability on failure trajectories.

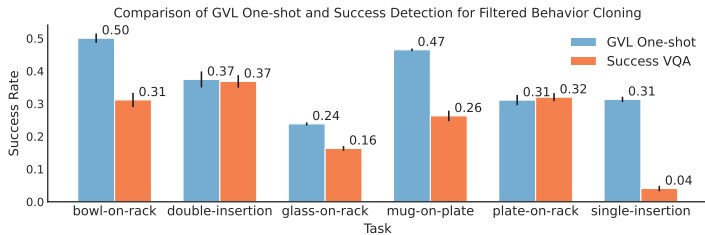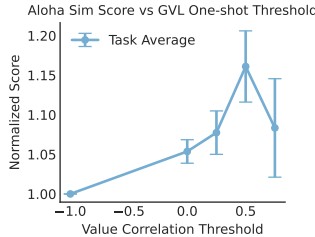

Figure 6: **Success-Filtered Imitation Learning on ALOHA Simulation Tasks.** Left: Using GVL-SD for success-filtered BC substantially outperforms SuccessVQA. Right: GVL-SD is not sensitive to the VOC threshold for improving imitation learning.

low precision, indicating that the base VLM systematically biases towards outputting failure. Adding one in-context demonstration further improves GVL's performance across all metrics.

In Fig. 5 (Left), we also visualize the histogram of the VOC scores GVL produces on success and failure trajectories. As expected, GVL on failure trajectories renders a uniform distribution when the task is unsuccessful, indicating the model's inability to uncover the original temporal order – success and failure trajectories have distinct distributions over the correlation values indicating that GVL can adequately separate them. Fig. 5 (Right) shows that the histograms without shuffling are largely the same independent of success or failure. This shows that by forcing the VLM to perform the more difficult prediction task over shuffled frames, GVL can elicit better zero-shot values.

Now, we use the above success detection methods for filtered imitation learning; for all methods, we use Action Chunking Transformer (ACT) as the imitation learning algorithm (Zhao et al., 2023); ACT hyperparameters are tuned for ACT on the success-only subset and are fixed for all methods. Given the noisiness in model checkpoints performance, we report the average success rate of the last 10 model training checkpoints. Results for the six simulation tasks are shown to the

| Real-World ALOHA Tasks | GVL + DP | DP | Avg. VOC |
|---|---|---|---|
| `bowl-in-rack` | **7/10** | 6/10 | 0.57 |
| `banana-handover` | **7/10** | 5/10 | 0.73 |
| `close-laptop` | **9/10** | 6.5/10 | 0.59 |
| `open-drawer` | 4/10 | **6/10** | 0.09 |
| `remove-gears` | 4.67/10 | **7/10** | 0.19 |
| `pen-handover` | **1.5/10** | 0/10 | 0.43 |
| `fold-dress` | 7/10 | 7/10 | 0.66 |

Table 3: **Real-World ALOHA Policy Learning Results.** AWR with GVL (One-Shot) outperforms imitation learning baselines when the value predictions have high VOCs.

right of Fig. 6, where GVL-SD's improved success detection leads to better performance over SuccessVQA. In fact, SuccessVQA often hurts performance, likely because of its low precision which causes the policy to train on a high number of false positive (i.e. failure) trajectories. In Fig. 6 (Right) we show the effect of varying the VOC threshold in $\{-1.0, 0, 0.25, 0.5, 0.75\}$ in comparison to training on all the data with ACT; note that this is the same using the lowest threshold, $-1.0$ as it is a lower bound on the VOC metric. As seen, GVL consistently outperforms ACT regardless of threshold values; when the threshold value is too high, i.e., $0.75$, we see a slight dip in performance when the overall dataset size becomes too small.

**Advantage-weighted regression for real-world visuomotor control.** Finally, we illustrate how GVL can assign importance weights to individual transitions within trajectories at a fine-grained

level akin to offline reinforcement learning. For these experiments we use real-world demonstration data collected by human teleoperation on bi-manual ALOHA robot setups. Unlike simulation, our datasets only contain successful task executions but can be sub-optimal and multi-modal. Thus, we directly utilize GVL's values with *advantage weighted regression* (AWR) (Peters & Schaal, 2007; Peng et al., 2019), in which we weight each individual transition by the estimated advantage, or GVL value difference for that step:

$$\mathcal{L}(\theta) := -\mathbb{E}\left[\exp\left(\tau \cdot (v_{k+1} - v_k)\right) \cdot \log \pi_\theta(a_k \mid o_k)\right] \tag{6}$$

We use diffusion policy (DP) as the policy backbone (Chi et al., 2023) for each task, and compare training diffusion policies with GVL (One-Shot) advantage weighting or lack thereof. We evaluate on 7 tasks with 10 trials per task and report success rate in Table 3. As can be seen, on a majority tasks, GVL-DP outperforms DP and we see a clear correlation between improvement over DP and the VOC score. That is, when the value predictions are of high quality as judged by VOC, policy learning can benefit from GVL value weighting. On `open-drawer` and `remove-gears`, the top-down view does not provide sufficient resolution to distinguish task progress (see Fig. 8), as a consequence, the value predictions can be noisy, which can hurt policy learning. However, given the in-context learning results, we believe that it is possible to improve policy learning even on difficult tasks with non-ideal camera viewpoints.

## 4.4 ABLATIONS

Finally, we ablate key algorithmic design choices of GVL to validate their necessity. In the Appendix, we additionally demonstrate that GVL's performance is robust to the choice of backbone VLMs as well as input camera viewpoint.

**Is autoregressive value prediction necessary?** We consider an ablation that simply asks the VLM to predict values of input observations one by one without GVL's autoregressive batch prediction mechanism. This ablation, which we refer to as **VLM (Single Frame)**, essentially poses value estimation as a VQA problem. In Tab. 4, we compare this ablation to GVL on a subset of RT-1 dataset as in Section 4.1; the average VOC for VLM (Single Frame) is a mere $-0.08$, a significant drop from GVL's 0.74 on RT-1 dataset. As seen, pre-trained VLMs by themselves are poor value estimators, generating inconsistent values that are too noisy to be used in practice.

**Is input observation shuffling necessary?** As discussed, we find that removing shuffling collapses ICV's predictions into generating degenerate values; that is, regardless of the quality of the provided trajectory, GVL tends to predict monotonically increasing values, resulting in inflated VOC scores that cannot be used to discriminate successful and failure trajectories.; see Fig. 5 (Right). To further qualitatively illustrate this phenomenon, in Fig. 11 in the Appendix, we overlay raw GVL value predictions with frame shuffling and

| RT-1 dataset | GVL | Single Frame |
|---|---|---|
| VOC | **0.74** | -0.08 |

Table 4: Using VLMs to predict values frame-by-frame significantly underperforms GVL's autoregressive mechanism.

lackthereof to understand the spread of the value curves. We see that the overlay for original GVL looks "messy", suggesting that GVL outputs varied value curves that better capture the heterogeneity of the queried video qualities. In contrast, without frame shuffling, GVL predictions indeed collapses onto a few linear ascending patterns.

## 5 CONCLUSION

We have introduced Generative Value Learning (GVL), a universal value function via VLM autoregressive value prediction on shuffled video frames. GVL can zero-shot output dense and high-quality value predictions for diverse and challenging real-world robotic tasks, spanning various robot embodiments and task categories. With few-shot learning from the same task, different task, or different embodiment, GVL performance steadily improves. We have demonstrated several use cases of using GVL to perform dataset, trajectory, and transition selection to improve downstream policy learning performance and generalization. We believe that GVL takes an important step in using foundation models supervision for robot learning.

ACKNOWLEDGMENT

We thank Jie Tan, Pannag Sanketi, Oliver Groth, and the rest the Google DeepMind Robotics team for helpful discussions and providing feedback on the paper. Yecheng Jason Ma is supported by the Apple Scholars in AI/ML PhD Fellowship and OpenAI Superalignment Fellowship. This project was supported in part by NSF CAREER 2239301, NSF Award 2331783, DARPA TIAMAT HR00112490421.

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

## A    LIMITATIONS AND FUTURE WORK

GVL is most suited for offline setting, in which full trajectories are available for inference. In the online setting, GVL must re-query each new sub-trajectory from every new step; this may be computationally expensive. Furthermore, we have not investigated whether pre-trained VLMs can be fine-tuned to perform better value predictions. In addition, though we test on diverse camera viewpoints, we have not yet investigated whether multi-view observations can improve value prediction quality. In addition, our evaluation metric Value-Order Correlation is most suitable for a-periodic tasks for which there exists a unique ordering of frames from an expert demonstration. Tasks such as wiping or stirring may be hard to discern. Though these limitations present avenues for future work, we believe GVL is a step towards improved in-the-wild value estimation.

## B    PROMPT

In this section, we provide the full prompt provided to the VLM for GVL predictions. The same prompt is used for all OXE datasets.

```
You are an expert roboticist tasked to predict task completion
    percentages for frames of a robot for the task of {
    task_description}. The task completion percentages are
    between 0 and 100, where 100 corresponds to full task
    completion. We provide several examples of the robot
    performing the task at various stages and their corresponding
     task completion percentages. Note that these frames are in
    random order, so please pay attention to the individual
    frames when reasoning about task completion percentage.

Initial robot scene:

In the initial robot scene, the task completion percentage is 0.

 Now, for the task of {task_description}, output the task
    completion percentage for the following frames that are
    presented in random order. For each frame, format your
    response as follow: Frame {i}: Frame Description: {}, Task
    Completion Percentages:{}%

Frame {i}:
```

## C    GVL OXE DATASET VOC BREAKDOWN

In this section, we provide the full list of average VOC score for each OXE dataset. In Table 5, we provide the VOC scores for GVL with `Gemini-1.5-Pro` as the backbone VLM. In Table 6, we provide the VOC scores for GVL with `GPT-4o` as the backbone VLM.

## D    SIMULATION TASKS

In Figure 7, we illustrate the six simulation tasks used for the success detection and filtered imitation learning experiment. For each task, we use VR teleoperation to collect 500 trajectories for initial policy training. After the policy converges, we rollout the last checkpoint for 1000 imtes, resulting in naturally balanced mix-quality datasets of about half success and half failure trajectories.

## E    REAL-WORLD TASKS

In Fig. 8, we provide the top-down view of the 7 ALOHA tasks used for real-world policy learning experiments.

| Dataset | VOC Score |
|---|---|
| utokyo_pr2_opening_fridge_converted_externally_to_rlds | 0.8095 |
| utokyo_xarm_bimanual_converted_externally_to_rlds | 0.7955 |
| utokyo_xarm_pick_and_place_converted_externally_to_rlds | 0.7880 |
| fractal20220817_data | 0.7385 |
| maniskill_dataset_converted_externally_to_rlds | 0.7260 |
| berkeley_autolab_ur5 | 0.7185 |
| nyu_door_opening_surprising_effectiveness | 0.6685 |
| utokyo_pr2_tabletop_manipulation_converted_externally_to_rlds | 0.5875 |
| utaustin_mutex | 0.5810 |
| iamlab_cmu_pickup_insert_converted_externally_to_rlds | 0.5585 |
| fmb | 0.5555 |
| ucsd_kitchen_dataset_converted_externally_to_rlds | 0.5295 |
| dobbe | 0.5295 |
| toto | 0.5270 |
| bridge | 0.5145 |
| austin_sirius_dataset_converted_externally_to_rlds | 0.5100 |
| asu_table_top_converted_externally_to_rlds | 0.5055 |
| berkeley_rpt_converted_externally_to_rlds | 0.4835 |
| berkeley_cable_routing | 0.4470 |
| usc_cloth_sim_converted_externally_to_rlds | 0.4410 |
| jaco_play | 0.4205 |
| bc_z | 0.4065 |
| viola | 0.4035 |
| berkeley_mvp_converted_externally_to_rlds | 0.3900 |
| roboturk | 0.3545 |
| austin_buds_dataset_converted_externally_to_rlds | 0.3415 |
| stanford_hydra_dataset_converted_externally_to_rlds | 0.3325 |
| tokyo_u_lsmo_converted_externally_to_rlds | 0.3140 |
| berkeley_fanuc_manipulation | 0.2685 |
| cmu_stretch | 0.2625 |
| ucsd_pick_and_place_dataset_converted_externally_to_rlds | 0.2410 |
| kuka | 0.1915 |
| dlr_sara_pour_converted_externally_to_rlds | 0.1600 |
| taco_play | 0.0945 |
| dlr_edan_shared_control_converted_externally_to_rlds | 0.0855 |
| droid | -0.0060 |
| stanford_robocook_converted_externally_to_rlds | -0.0690 |
| imperialcollege_sawyer_wrist_cam | -0.1225 |
| kaist_nonprehensile_converted_externally_to_rlds | -0.1310 |
| austin_sailor_dataset_converted_externally_to_rlds | -0.1715 |
| cmu_play_fusion | -0.3445 |
| stanford_kuka_multimodal_dataset_converted_externally_to_rlds | -0.3770 |
| stanford_mask_vit_converted_externally_to_rlds | -0.4505 |
| nyu_franka_play_dataset_converted_externally_to_rlds | -0.4555 |
| uiuc_d3field | -0.7025 |
| cmu_franka_exploration_dataset_converted_externally_to_rlds | -0.7395 |
| columbia_cairlab_pusht_real | -0.7625 |
| robo_net | -0.8485 |
| dlr_sara_grid_clamp_converted_externally_to_rlds | -1.0000 |

Table 5: GVL (`Gemini-1.5-Pro`) OXE Dataset VOC Scores

| Dataset | VOC Score |
|---|---|
| nyu_door_opening_surprising_effectiveness | 0.883 |
| utokyo_pr2_opening_fridge_converted_externally_to_rlds | 0.864 |
| berkeley_mvp_converted_externally_to_rlds | 0.8285 |
| utaustin_mutex | 0.813 |
| fractal20220817_data | 0.803 |
| utokyo_xarm_pick_and_place_converted_externally_to_rlds | 0.7665 |
| berkeley_autolab_ur5 | 0.755 |
| utokyo_xarm_bimanual_converted_externally_to_rlds | 0.749 |
| utokyo_pr2_tabletop_manipulation_converted_externally_to_rlds | 0.734 |
| austin_sirius_dataset_converted_externally_to_rlds | 0.7235 |
| toto | 0.713 |
| dlr_edan_shared_control_converted_externally_to_rlds | 0.6595 |
| bridge | 0.6445 |
| berkeley_fanuc_manipulation | 0.6295 |
| berkeley_rpt_converted_externally_to_rlds | 0.6235 |
| ucsd_kitchen_dataset_converted_externally_to_rlds | 0.603 |
| roboturk | 0.57 |
| jaco_play | 0.5615 |
| iamlab_cmu_pickup_insert_converted_externally_to_rlds | 0.557 |
| uiuc_d3field | 0.5395 |
| usc_cloth_sim_converted_externally_to_rlds | 0.5355 |
| asu_table_top_converted_externally_to_rlds | 0.5025 |
| maniskill_dataset_converted_externally_to_rlds | 0.499 |
| kaist_nonprehensile_converted_externally_to_rlds | 0.492 |
| viola | 0.4605 |
| austin_buds_dataset_converted_externally_to_rlds | 0.454 |
| cmu_play_fusion | 0.4235 |
| tokyo_u_lsmo_converted_externally_to_rlds | 0.3875 |
| austin_sailor_dataset_converted_externally_to_rlds | 0.3015 |
| ucsd_pick_and_place_dataset_converted_externally_to_rlds | 0.2675 |
| berkeley_cable_routing | 0.255 |
| dlr_sara_pour_converted_externally_to_rlds | 0.252 |
| imperialcollege_sawyer_wrist_cam | 0.239 |
| robo_net | 0.237 |
| stanford_hydra_dataset_converted_externally_to_rlds | 0.205 |
| cmu_stretch | 0.1895 |
| bc_z | 0.176 |
| nyu_franka_play_dataset_converted_externally_to_rlds | 0.1735 |
| stanford_robocook_converted_externally_to_rlds | 0.16 |
| kuka | 0.132 |
| stanford_mask_vit_converted_externally_to_rlds | -0.173 |
| stanford_kuka_multimodal_dataset_converted_externally_to_rlds | -0.1785 |
| columbia_cairlab_pusht_real | -0.1815 |
| cmu_franka_exploration_dataset_converted_externally_to_rlds | -0.2075 |
| taco_play | -0.2705 |
| eth_agent_affordances | -0.279 |
| dlr_sara_grid_clamp_converted_externally_to_rlds | -1 |

Table 6: GVL (`GPT-4o`) OXE Dataset VOC Scores

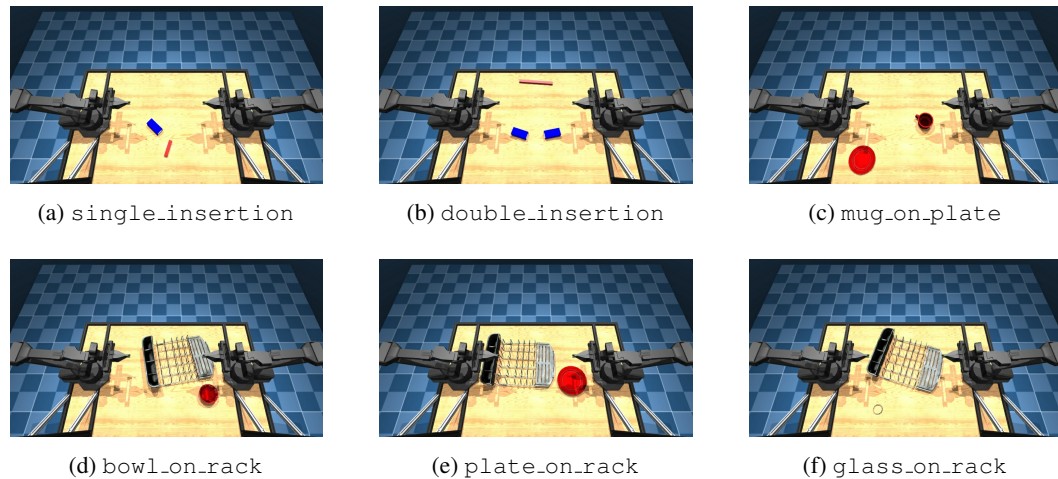

(a) `single_insertion`    (b) `double_insertion`    (c) `mug_on_plate`

(d) `bowl_on_rack`    (e) `plate_on_rack`    (f) `glass_on_rack`

Figure 7: Simulated task setups of dexterous manipulation on the ALOHA robot.

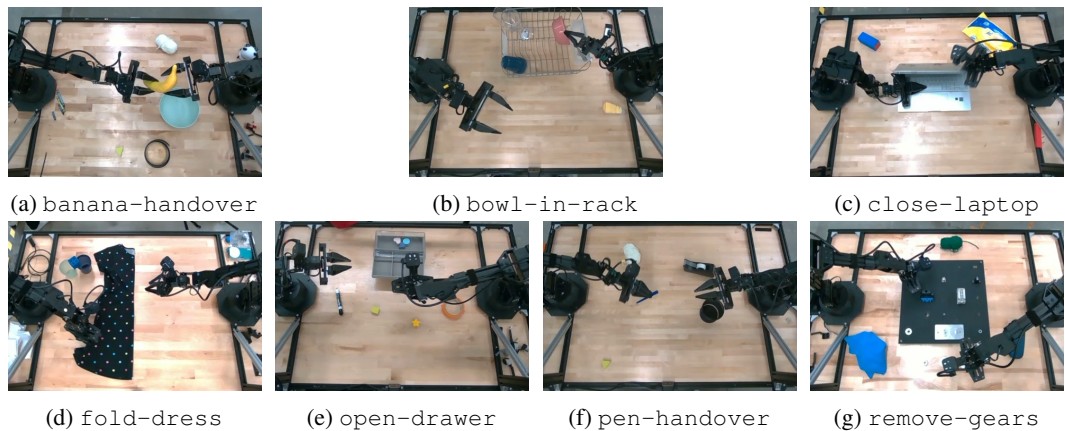

(a) `banana-handover`    (b) `bowl-in-rack`    (c) `close-laptop`

(d) `fold-dress`    (e) `open-drawer`    (f) `pen-handover`    (g) `remove-gears`

Figure 8: Real-world experiment setups for dexterous manipulation on the ALOHA robot.

## F GVL VISUALIZATION

In this section, we visualize several raw GVL predictions on our ALOHA datasets. Samples are chosen from diverse camera viewpoints for which we evaluated GVL. The visualizations are in Fig. 9. As shown, on diverse viewpoints, GVL remains effective for diverse tasks. We do observe that GVL sometimes would predict "spike" values that are not consistent with the rest of the predictions. We hypothesize that this could be due to partial observability of the task at that particular timestep as well as inherent stochasticity in the shuffling order; we leave to future work for a more systematic investigation of these types of errors.

## G ADDITIONAL RESULTS

In this section, we present additional results and analysis.

**GVL and LIV comparison on a subset of high-quality OXE datasets.** Given that not all OXE datasets are necessarily demonstration data most suitable for being used for VOC-based evaluation, we have delegate a few datasets to be of "high-quality" to use VOC as a metric and re-create the original results as in Fig. 2. To this end, we have selected RT-1 (Brohan et al., 2023), Bridge (Ebert et al., 2021; Walke et al., 2023),DOBBE (Shafiullah et al., 2023), and BC-Z (Jang et al., 2022) datasets because these datasets have successfully been used in prior works for large-scale imitation learning and include diverse tasks and camera viewpoints. The results are illustrated in Fig. 10. As

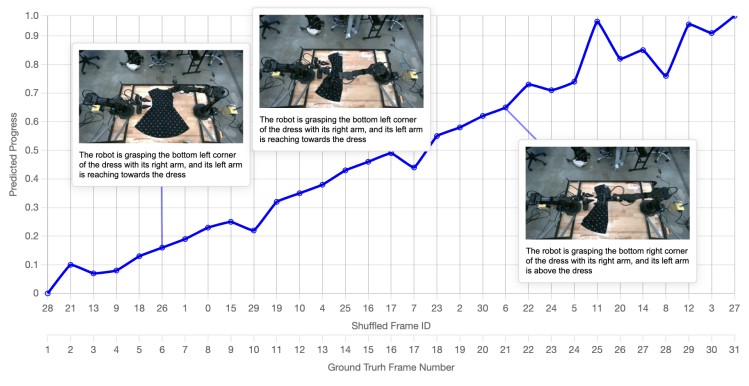

(a) `fold_dress` from the top-down view.

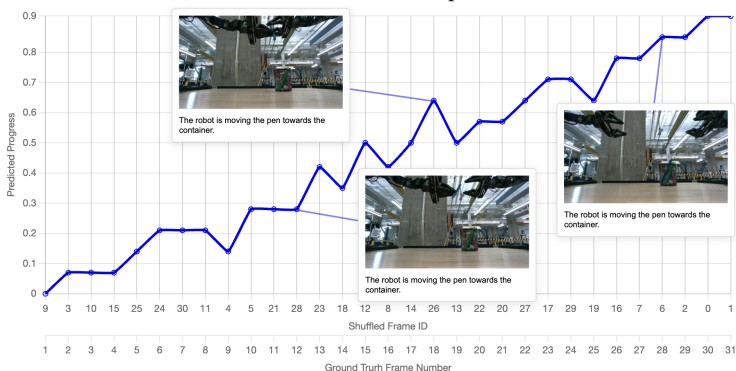

(b) `pen_handover` from the table view.

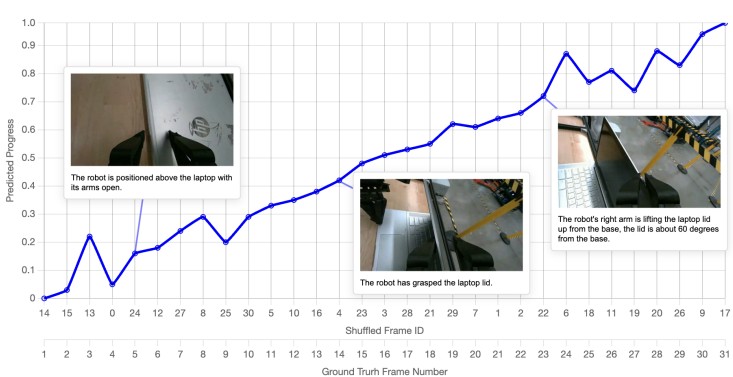

(c) `close_laptop` from the right wrist camera.

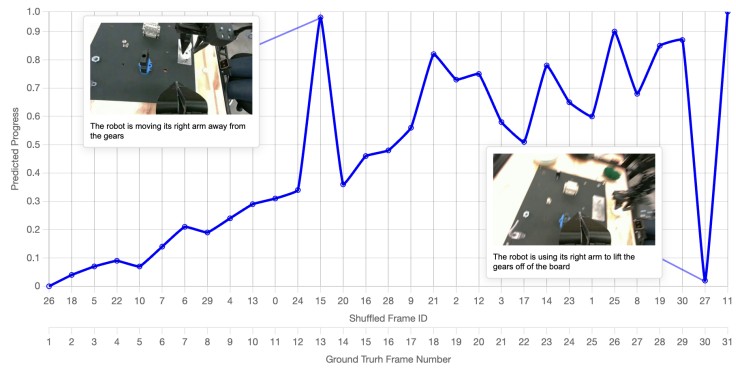

(d) `remove_gears` from the left wrist camera.

Figure 9: Example GVL predictions on diverse ALOHA tasks.

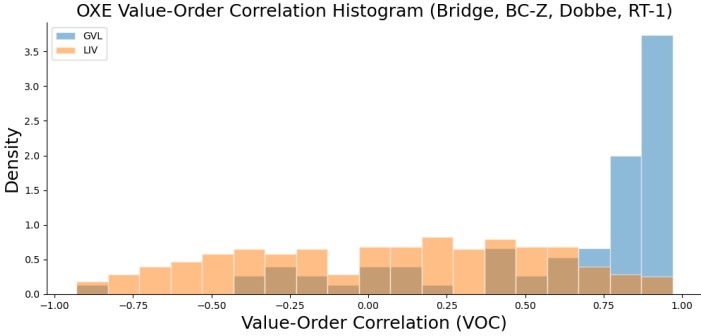

Figure 10: On a subset of high-quality OXE datasets, GVL consistently outperforms LIV, mirroring the comprehensive OXE results in Fig. 2.

seen, on this subset of high-quality and diverse OXE datasets, GVL consistently generates highly positive VOCs; in contrast, LIV generates VOCs close to a uniform distribution, indicating that it struggles to accurately predict language-conditioned task progress on unseen robot videos.

**GVL's qualitative behavior on task failure and repetition.** In our supplementary material submission, we have included two videos in which repetition or failure is present. As shown, in both cases, when the robot gripper pulls back from the object of task interest, GVL's task progress estimates decrease, and when the gripper recovers and makes progress again, GVL's estimates increase. These results demonstrate GVL's ability to accurately estimate values in videos when repetition is present.

**GVL and No-Shuffling ablation qualitative comparison.** As shown in Fig. 11, GVL generates value predictions that are varied over time; in contrast, without frame shuffling, the predictions all collapses onto a few monotonic patterns.

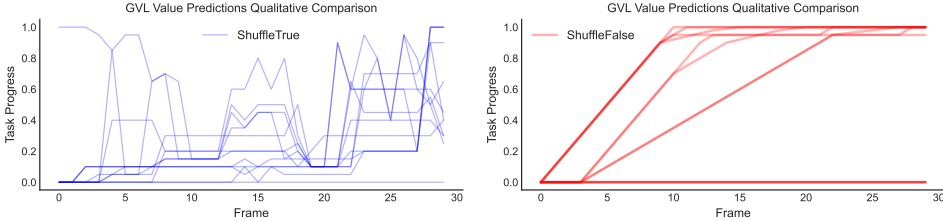

Figure 11: GVL without shuffling produces uninformative monotonic values regardless of trajectory quality.

**DROID co-training dataset filtering results.** We use GVL to filter a subset of random demonstrations from the DROID dataset (Khazatsky et al., 2024). We first use GVL to construct VOC scores for each of the two external cameras for 1243 demonstrations. Then, we filter the demonstrations according to the average VOC score across both camera images. As policies trained on DROID do not perform well zero-shot, we follow prior works (Khazatsky et al., 2024; Hejna et al., 2024) and co-train policies with a handful of in-domain demonstrations to asses the quality of the filtered data. We include 15 demonstrations for each of three different tasks: 1) "Purple Bowl", where the robot places a purple bowl in a dish rack, 2) "Blue Pot" where the robot places a blue pot in the dish rack, and 3) a more challenging "Bowl Flip", where the robot has to first flip over a large bowl, and then place it in the dish rack upside down. We train Diffusion policies (Chi et al., 2023) with a data mixture comprised of 60% demonstrations from DROID, filtered according to different methods, and 40% 15 in-domain demonstrations for each task. Full results can be found in Table 7. Similar to Khazatsky et al. (2024), we find that co-training with more data helps performance. GVL notably attains the same performance training with only 947 demonstrations as training on the entire subset of 1243 demonstrations. However, training on a random subset of 947 demonstrations incurs a performance hit – reducing average success rate from 20/40 to only 7/40. All policies generally struggle more with the harder flipping task, especially with the blue bowl.

| Method | # Demos | Purple Bowl | Blue Pot | Flip Bowl - Orange | Flip Bowl - Blue | Total |
|--------|---------|-------------|----------|--------------------|-------------------|-------|
| All Data | 1243 | 9/10 | 6/10 | 4/10 | 1/10 | 20/40 |
| Random Subset | 947 | 2/10 | 2/10 | 2/10 | 1/10 | 7/40 |
| GVL, VOC 0.0 | 947 | 10/10 | 6/10 | 3/10 | 1/10 | 20/40 |

Table 7: Policy Learning results using GVL to filter the droid dataset.

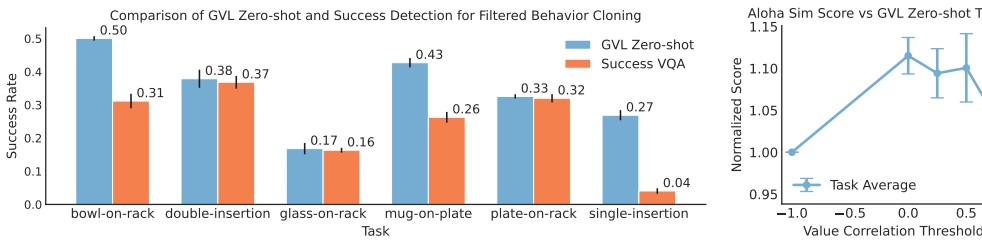

Figure 12: **Success-Filtered Imitation Learning on ALOHA Tasks.** Left: Using GVL-SD for success-filtered BC substantially outperforms SuccessVQA. Right: GVL-SD is not sensitive to the VOC threshold for improving imitation learning.

**Zero-Shot Aloha Sim Results.** In Fig. 12 we include results for GVL-SD zero-shot instead of one-shot. The results are qualitatively similar, where GVL-SD consistently outperforms SuccessVQA, and different VOC threshold values all provide performance gain.

**Different VLM backbone.** We additionally consider GPT-4o as the backbone VLM to better understand GVL's performance in relation to the backbone VLM model. For evaluation, we plot the histogram of all 1000 (50×20) Value Order Correlation (VOC) scores across all trajectories in Figure 13. As shown, GVL, independent of the backbone models, consistently generates VOC scores that heavily skews to the right, indicating that it is able to zero-shot recover the temporal structure hidden in the shuffled demonstration videos, i.e., coherent value predictions.

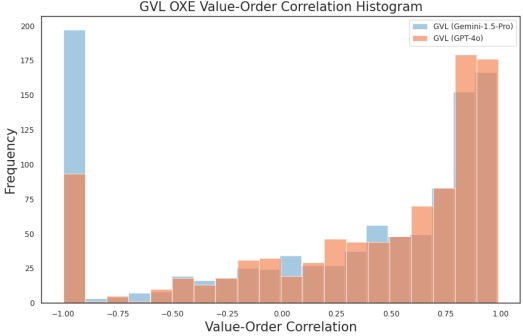

Figure 13: GVL has comparable performances with different backbone VLMs; the main difference is in the backbone model's refusal rate and conforming to the response template, which is reflected in the tall bar at −1.0.

**Cross-task in-context learning.** We investigate whether examples from other tasks can also unlock GVL's in-context learning capability. On the previous ALOHA-13 tasks, we randomly pair up tasks, where we draw one demonstration from one task as the one-shot in-context example for another. Then, we compare VOCs with the original same-task one-shot setup. The results are shown in Figure 14. We see that providing examples from a different task is still beneficial, though the improvement is not as much as same-task examples. This is to be expected as intra-task examples still provide clue on the output format as well as a generic notion of task progress, but such information is not specific to the target task. That said, cross-task ICL enables the flexibility of enabling foundation model guidance on a task without any task-specific prior.

**Does GVL work on different camera viewpoints?** On our ALOHA setup, we collected all demonstrations using four camera viewpoints. Besides the top-down view reported in the main experiment above, we test whether GVL remains performant when using alternative viewpoints, especially

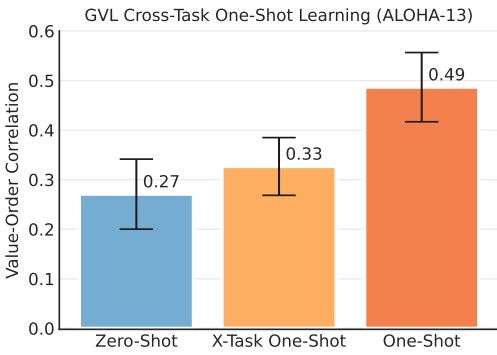

Figure 14: GVL demonstrates cross-task in-context learning capability: its value predictions can be improved by value examples from different tasks.

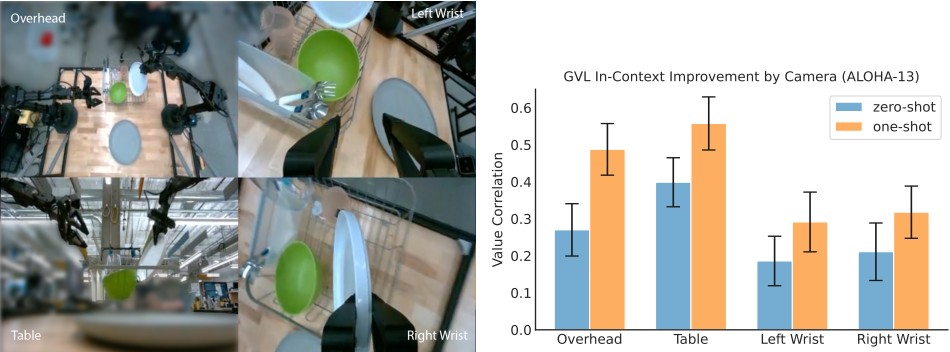

Figure 15: GVL works better on more in-distribution table view, but one-shot improvement benefits all camera views.

gripper views that are likely more out-of-distribution with respect to the natural images used for VLM training. The aggregate zero-shot and one-shot results are shown in Figure 15. As seen, on average, GVL zero-shot works best on the Table viewpoint. This is not surprising, as images taken with the front facing table camera are arguably visually closer to naturally captured images used for VLM training. Yet, with in-context examples, GVL consistently improves on all camera viewpoints. In practice, this means that GVL is robust to camera viewpoints – even when a camera viewpoint is determined to be sub-optimal post-hoc, practitioners can make up for that by simply providing few in-context examples.

**Does GVL pay attention to the task specification?** To validate that GVL is not merely recovering the temporal coherence in the shuffled input video but actively tracking visual progress according to the task language command, we compute the VOC scores for every combination of task input video and language description in the ALOHA-13 split. The heatmap visualization of the average VOC for every pairing is illustrated in Fig. 16 and Fig. 17 for GVL and the no shuffling ablation. On 9 out of 13 tasks, GVL achieves the highest VOC when the input video and the task description matches; in many unmatched cases, the model simply refuses to output value predictions, stating that the frames and the language description are not related. In contrast, when we do not shuffle the input frames, the quality dramatically drops.

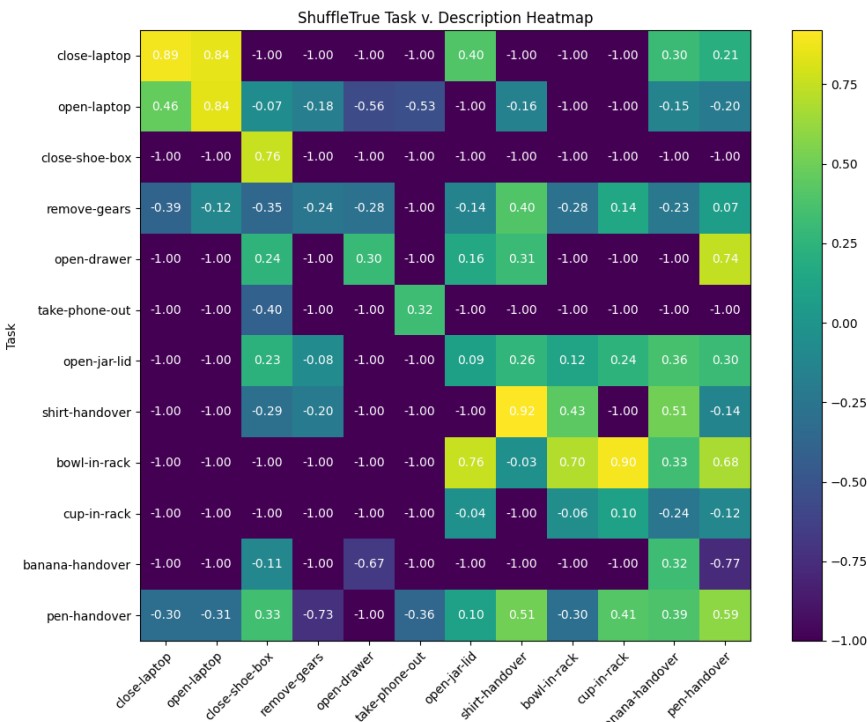

Figure 16: GVL VOC for video and language description pairs. Shuffling enables GVL to pay attention to the language task description in order to faithfully predict observation values.

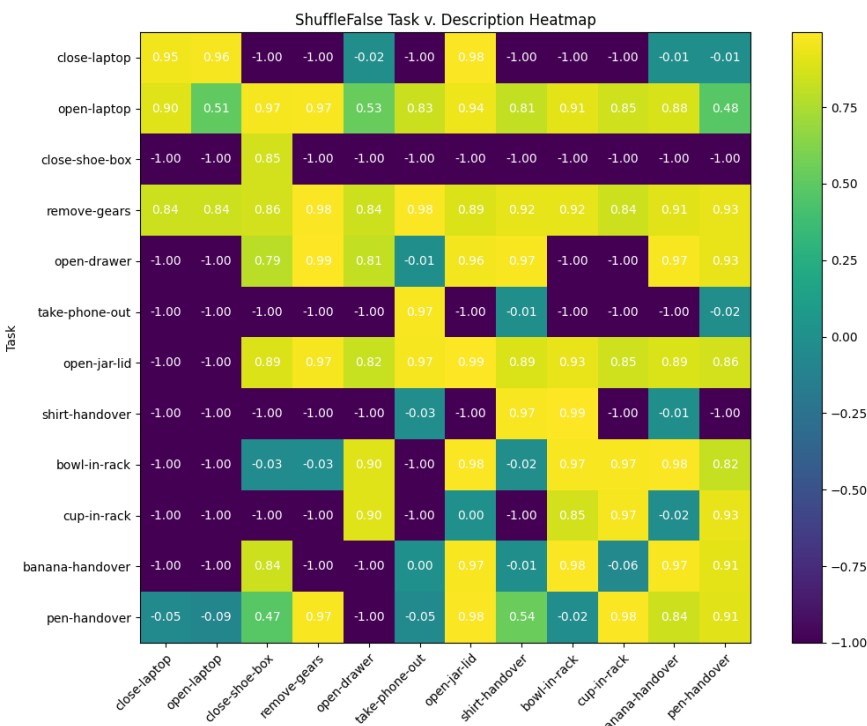

Figure 17: No-shuffling ablation VOC for video and language description pairs. Removing shuffling makes VLM output high VOCs independent of task descriptions.

