# OpenReview forum: "Vision Language Models are In-Context Value Learners"
_ICLR.cc/2025/Conference — ICLR 2025 Spotlight_

### Official Review · Reviewer_nm81 · 2024-10-31

**Soundness:** 4
**Presentation:** 3
**Contribution:** 3
**Rating:** 8
**Confidence:** 3

**Summary:**

This paper casts universal value estimation of video frames as a temporal ordering problem over shuffled video frames. To address this, the authors introduce Generative Value Learning (GVL), which leverages the world knowledge embedded in vision-language models (VLMs) to estimate task progress. Here, the value for each frame is represented as the VLM's prediction of task completion rate. Experimental results demonstrate that GVL can be effectively applied to various downstream tasks, using a Value-Order Correlation (VOC) metric that measures the rank correlation between the rank of estimated values and the original frame order.

Overall, I believe this paper is suitable for publication at ICLR, though there is room for further development, especially in the clarity of the writing.

**Strengths:**

- Framing value estimation as a temporal ordering problem over shuffled videos is a compelling approach.
- The proposed rank correlation metric is thoughtfully designed and well-suited to this problem setting.
- The experiments are extensive, covering a wide range of real-world tasks.

**Weaknesses:**

- My primary concern is with the method's applicability. As I understand, calculating the value for each frame requires access to a complete video, which could limit its use in online policy learning without an offline dataset or expert demonstrations. This restricts the downstream tasks to scenarios where offline datasets are available, while the compared method [1] remains applicable in online policy learning.
- The manuscript contains several typos and minor errors that should be addressed. I recommend a careful proofreading. A non-exhaustive list includes:
    - Line 163: missing comma.
    - Line 178: missing space after “2).”
    - Line 197: "meaningfully" should be "meaningful."
    - Line 393: "ataset" should be "dataset."

[1] Yecheng Jason Ma, Vikash Kumar, Amy Zhang, Osbert Bastani, and Dinesh Jayaraman. Liv: Language-image representations and rewards for robotic control. In International Conference on Machine Learning, pp. 23301–23320. PMLR, 2023.

**Questions:**

- In Figure 1, some arrows are bidirectional. Replacing these with unidirectional arrows could improve consistency.
- In the Dataset Quality Estimation paragraph of Section 4.3, the reasoning feels incomplete. It may strengthen the experiment to directly train models with and without specific low-quality datasets and compare the resulting performance outcomes.

---

> ### Author Response · Authors · 2024-11-22
>
> Dear reviewer nm81,
>
> Thank you for your thoughtful comments and feedback! Here, we respond to specific questions and concerns the reviewer raises.
>
> ---------------------------------------
> **Question/Comment 1**: Method’s applicability to online RL.
>
> **Response 1**:   Yes, GVL is applicable to online RL. Many commonly used online RL algorithms, such as SAC [1], maintain a replay buffer for trajectories the policy has encountered during exploration. Recent practical methods for online RL also bootstrap from offline datasets (e.g., [2, 3]). In all these settings, GVL can be applied the same way. However, we do note that in a pure online setting where we need value estimates for every step as soon as the step is executed, then we would need to continuously apply GVL to partial trajectories, which can be slow and expensive. But we believe that this may not be a practical concern given the limited use cases of such pure online settings in practical robotics applications. Nevertheless, we have added a discussion on this topic in our limitation and discussion section.
>
> [1] Haarnoja, Tuomas, et al. "Soft actor-critic: Off-policy maximum entropy deep reinforcement learning with a stochastic actor." International conference on machine learning. PMLR, 2018.
> [2] Ball, Philip J., et al. "Efficient online reinforcement learning with offline data." International Conference on Machine Learning. PMLR, 2023.
> [3]  Nakamoto, Mitsuhiko, et al. "Cal-ql: Calibrated offline rl pre-training for efficient online fine-tuning." Advances in Neural Information Processing Systems 36 (2024).
>
>
>
>
>
> ---------------------------------------
> **Question/Comment 2**: Dataset Quality Estimation Section 4.3 can be strengthened with empirical results.
>
> **Response 2**: Thanks for this suggestion! We have added one more real-world experiment during the rebuttal period. Details are in Appendix F of the updated manuscript. Given that we find DROID to be a dataset that has low VOCs in Section 4.3, we investigate whether this finding allows us to train better policies in the setting of co-training on the DROID dataset, which was how the dataset was used for real-world imitation learning in the original paper. Specifically, we study whether GVL’s data quality estimation capability can be used to discover better co-training datasets from a raw, unfiltered diverse dataset like DROID. In Table 7 in Appendix F, we find that using GVL filtering can enable diffusion policies to (1) match the performance when co-trained on bigger datasets while converging faster, and (2) significantly outperform a baseline where the co-training dataset size is kept the same as the GVL filtered dataset but randomly selected.
>
>
>
> ---------------------------------------
> **Question/Comment3**: Figure 1 and typos should be fixed.
>
> **Response 3**: Thanks for catching these issues. We have proofread and revised our manuscript to improve its presentation.
>
>
>
> We thank the reviewer again for their time and effort helping us improve our paper! Please let us know if we can provide additional clarifications to improve our score.

---

> > ### Comment · Reviewer_nm81 · 2024-11-25
> >
> > I appreciate the authors for thoroughly addressing all of my concerns. I am satisfied with their response and am willing to raise my score.

---

### Official Review · Reviewer_tm1A · 2024-11-02

**Soundness:** 3
**Presentation:** 4
**Contribution:** 3
**Rating:** 8
**Confidence:** 4

**Summary:**

This paper presents Generative Value Learning (GVL) a value function estimator for predicting task progress from visual trajectories. They show that without any task-specific training, their approach can in-context zero-shot and few-shot predict values for more than 300 real-world tasks across different embodiments. The approach is simple: autoregressive prediction, shuffle inputs, and provide in-context examples of task progress.

**Strengths:**

- Well motivated and set desideratas for a “universal value estimator”: accurately estimating state and consistency, highlighting the limitations of prior works
- Interesting finding; shuffling the video frames help mitigate the temporal bias found in videos ⇒ better value estimations
- Introduces a new evaluation metric: Value-Order Correlation which measures how well predicted values correlate with the ground-truth timestep in expert videos.
- Extensive evaluation of GVL’s value estimates across a large suite of real-world robot datasets with varying tasks and embodiments
- GVL can be applied to a wide variety of tasks such as dataset quality estimation, success detection, and weighted imitation learning.
- Good analysis and ablation studies to verify the importance of each component in the GVL approach and robustness to different environment factors.

**Weaknesses:**

- The real-world results are not too impressive. Applying GVL to compute values only leads to a small improvement over the based diffusion policy baseline. It would be interesting to see more results showing how the value estimations of GVL help with downstream policy learning.
- Method itself is technically simple, but still well-motivated and supported with empirical results.

**Questions:**

- Can GVL be applied for online RL training?

---

> ### Author Response · Authors · 2024-11-22
>
> Dear reviewer tm1A,
>
> Thank you for your thoughtful comments and feedback! Here, we respond to specific questions and concerns the reviewer raises.
>
> ---------------------------------------
> **Question/Comment 1**: More real-world results showing how the value estimations of GVL help with downstream policy learning can strengthen the experiment section.
>
> **Response 1**: Thanks for this suggestion! We have added one more real-world experiment during the rebuttal period. Details are in Appendix F of the updated manuscript. Given that we find DROID to be a dataset that has low VOCs in Section 4.3, we investigate whether this finding allows us to train better policies in the setting of co-training on the DROID dataset, which was how the dataset was used for real-world imitation learning in the original paper. Specifically, we study whether GVL’s data quality estimation capability can be used to discover better co-training datasets from a raw, unfiltered diverse dataset like DROID. In Table 7 in Appendix F, we find that using GVL filtering can enable diffusion policies to (1) match the performance when co-trained on bigger datasets while converging faster, and (2) significantly outperform a baseline where the co-training dataset size is kept the same as the GVL filtered dataset but randomly selected.
>
>
> ---------------------------------------
> **Question/Comment 2**: Is GVL applicable to online RL?
>
> **Response 2**: Yes, GVL is applicable to online RL. Many commonly used online RL algorithms, such as SAC [1], maintain a replay buffer for trajectories the policy has encountered during exploration. Recent practical methods for online RL also bootstrap from offline datasets (e.g., [2, 3]). In all these settings, GVL can be applied the same way. However, we do note that in a pure online setting where we need value estimates for every step as soon as the step is executed, then we would need to continuously apply GVL to partial trajectories, which can be slow and expensive. But we believe that this may not be a practical concern given the limited use cases of such pure online settings in practical robotics applications. Nevertheless, we have added a discussion on this topic in our limitation and discussion section.
>
> [1] Haarnoja, Tuomas, et al. "Soft actor-critic: Off-policy maximum entropy deep reinforcement learning with a stochastic actor." International conference on machine learning. PMLR, 2018.
> [2] Ball, Philip J., et al. "Efficient online reinforcement learning with offline data." International Conference on Machine Learning. PMLR, 2023.
> [3]  Nakamoto, Mitsuhiko, et al. "Cal-ql: Calibrated offline rl pre-training for efficient online fine-tuning." Advances in Neural Information Processing Systems 36 (2024).
>
> We thank the reviewer again for their time and effort helping us improve our paper! Please let us know if we can provide additional clarifications to improve our score.

---

> > ### Comment · Reviewer_tm1A · 2024-11-25
> >
> > I appreciate the author's response and additional discussion / real-world experiments. However, I will maintain my current score.

---

### Official Review · Reviewer_1PKJ · 2024-11-03

**Soundness:** 3
**Presentation:** 3
**Contribution:** 3
**Rating:** 6
**Confidence:** 3

**Summary:**

The paper presents a method that enables existing VLMs to understand the task progress/completion, that could help in the use cases like dataset filtering, success detection and visuomotor control/policy learning. The authors also propose an autoregressive value prediction instead of the traditional bellman returns.

**Strengths:**

Pros:
1. The high-level of the paper is easy to understand because of the effort that the authors took to portray it on the figures.
2. The experiments that were performed to evaluate the method are easy to see how the proposed method is better than the baseline/(baselines in some cases).

**Weaknesses:**

Weakness
1. Currently, the block diagram only provides a high-level idea of what the paper is about, which is good. But it would to nice to have the language component of the model. For example, I get that you are trying to predict the value estimate of a frame, but without the task description, it’d be impossible to predict whats the value estimate.
2. The experiments are very limited, although the use cases are diverse enough. In each of these use cases, however, the comparisons are quite restricted. My understanding is that you are using a policy that is learnt on the downstream task. There are so many representation learning works that do that and I urge the authors to find more recent baselines for the use cases, especially, for the visuomotor control
3. Why are the authors not comparing with LIV on the visuomotor tasks?
4. Is there a plot/table that shows how much incorporating observation shuffling helps or using autoregressive value prediction helps than the traditional bellman returns? I think this is important to understand the efficacies of the 2 proposed incorporations

**Questions:**

Overall, I enjoyed reading the paper, but I'm just concerned with the strength of the experiment section. I'm willing to consider changing my score if the experiments section is made stronger.

---

> ### Author Response · Authors · 2024-11-22
>
> Dear reviewer 1PKJ,
>
> Thank you for your thoughtful comments and feedback! Here, we respond to specific questions and concerns the reviewer raises.
>
> ---------------------------------------
> **Question/Comment 1**: The block diagram should include a language block.
>
> **Response 1**: Thank you for this suggestion. We have revised Figure 1 accordingly in our updated manuscript. Please let us know whether the clarity of our concept diagram can be further improved.
>
>
> ---------------------------------------
> **Question/Comment 2**: Comparison against representation learning methods, including LIV, is missing for visuomotor control tasks.
>
> **Response 2**: We’d like to clarify that the purpose of our visuomotor control experiments is to compare different dataset compositions generated by GVL and lack thereof, instead of comparing different choices of pre-trained visual representations. In all our visuomotor control experiments, we fix the choice of initial representations and believe our experimental findings hold independent of that choice. Specifically, we use state-of-art algorithms such as Action-Chunking Transformers (ACT) [1] or Diffusion Policy (DP) [2], and compare variants that differ in the training datasets; the vision encoders for both algorithms are Resnet-18, and they are continuously fine-tuned during imitation learning. Please let us know whether you have any remaining questions on this topic, and we are happy to provide additional clarifications during the rebuttal period.
>
> [1] Zhao, Tony Z., et al. "Learning fine-grained bimanual manipulation with low-cost hardware." arXiv preprint arXiv:2304.13705 (2023).
> [2] Chi, Cheng, et al. "Diffusion policy: Visuomotor policy learning via action diffusion." The International Journal of Robotics Research (2023): 02783649241273668.
>
>
> ---------------------------------------
> **Question/Comment 3**: Additional experiments can strengthen the paper.
>
> **Response 3**: Thanks for this suggestion! We have added one more real-world experiment during the rebuttal period; details are in Appendix F of the updated manuscript. Given that we find DROID to be a dataset that has low VOCs in Section 4.3, we investigate whether this finding allows us to train better policies in the setting of co-training on the DROID dataset, which was how the DROID dataset was used for real-world imitation learning in the original paper. Specifically, we study whether GVL’s data quality estimation capability can be used to discover better co-training datasets from a raw, unfiltered diverse dataset like DROID. In Table 7 in Appendix F, we find that using GVL filtering can enable diffusion policies to (1) match the performance when co-trained on bigger datasets while converging faster, and (2) significantly outperform a baseline where the co-training dataset size is kept the same as the GVL filtered dataset but randomly selected.
>
>
> ---------------------------------------
> **Question/Comment 4**: Is there a plot/table that shows how much incorporating observation shuffling helps or using autoregressive value prediction helps than the traditional bellman returns? I think this is important to understand the efficacies of the 2 proposed incorporations
>
> **Response 4**:
>
> Yes, we have compared to the ablation that does not incorporate autoregressive value prediction in Section 4.4; in our updated manuscript, Table 4 displays the result. In short, we find removing autoregressive value predictions substantially hamper VLMs’ value prediction capability, reducing it to a random level performance; this is precisely due to the fact that VLMs can no longer maintain any consistency among its task progress predictions.
>
> Comparison to no-shuffling ablation is presented in Figure 5, and again qualitatively in Figure 10 in Appendix; we find that removing shuffling makes the model insensitive to the quality of the input video and output monotonic values even when the video fails at solving the language task description.
>
> In terms of comparison to traditional bellman returns, we have compared to LIV [1], which uses a variant of the traditional bellman return to learn value functions. We have shown that GVL significantly outperforms LIV on all evaluation datasets. Note that we are not able to fine-tune a pre-trained VLM, such as Gemini or GPT-4o, using bellman-based objectives; however, given that GVL is able to attain satisfactory results without doing any fine-tuning, we see that as an advantage of the proposed approach.
>
> [1] Ma, Yecheng Jason, et al. "Liv: Language-image representations and rewards for robotic control." International Conference on Machine Learning. PMLR, 2023.
>
> We thank the reviewer again for their time and effort helping us improve our paper! Please let us know if we can provide additional clarifications to improve our score.

---

> > ### Comment · Reviewer_1PKJ · 2024-11-25
> > **Official reply**
> >
> > Thanks for adding the additional results and answering my questions. I've decided to increase my score from 5 to 6.

---

### Official Review · Reviewer_RQdG · 2024-11-04

**Soundness:** 3
**Presentation:** 4
**Contribution:** 3
**Rating:** 8
**Confidence:** 3

**Summary:**

This paper presents Generative Value Learning (GVL), a novel technique that utilizes Vision-Language Models (VLMs) to serve as a value function. In particular, by shuffling the order of frames within the video and utilizing few-shot prompting, GVL exceeds the performance achievable using VLMs without shuffling and without temporal dependence. Furthermore, the authors demonstrate that, by comparing the GVL value estimates with the true frame order, it is possible to identify and filter out low quality video demonstration data. The authors demonstrate that GVL can be used to train more effective policies via both data filtering and more accurate value estimation.

**Strengths:**

Although the proposed technique is simple, it seems (based on myriad experiments) to be effective. Figure 2 demonstrates a clear superiority of GVL over LIV, the baseline that the authors chose to test, for text-based goals. Figure 4 demonstrates that GVL can be decently effective at predicting values correctly over more complex videos. Finally, GVL was effective at training agents via both (1) data filtering for imitation learning and (2) advantage estimation for advantage weighted regression.

Breaking things down by category:
- Originality: Due to the simplicity of this technique, I think the originality is somewhat low, however the method is effective, especially in comparison with other simple alternatives.
- Quality: With the exception of the potential circularity I mention below, the experiments performed seemed to be of high quality, and they seem to demonstrate the quality of this technique.
- Clarity: There were some minor questions that I have detailed below, but for the most part, the paper was well-written and easy to understand.
- Significance: The authors provide solid motivation for the significance of this technique by demonstrating how it could be used for model training and dataset quality checks.

**Weaknesses:**

The use of Value Order Correlation (VOC)—the correlation between GVL and frame order—seemed somewhat circular. It is used on the OXE dataset to judge GVL’s performance, a choice that assumes that the contents of OXE are of sufficient quality. (If, for instance, OXE contained many sub-optimal trajectories, an ideal value estimator would show low VOC.) Later, there is a discussion about the use of GVL+VOC to identify sub-optimal datapoints/subsets within OXE. This seemed confusing to me since it seemed to contradict the earlier assertion that OXE was of sufficient quality for judging GVL.

It would be good to see more analysis regarding GVL’s ability to accurately estimate values in videos where repetition is present. Currently, the existence of such an ability has to be inferred from downstream effects (e.g., the ability to use GVL for data filtering helps during imitation learning).

**Questions:**

- It would be great to see LIV’s performance on the ALOHA dataset. I assume that, since GVL outperformed LIV on OXE, it would do so again on ALOHA, but it would be nice to see these results formalized.
- What is ALOHA-13? I see the descriptions of ALOHA-250, but I don’t see the same for ALOHA-13 for some reason.

---

> ### Author Response · Authors · 2024-11-22
>
> Dear reviewer RQdG,
>
> Thank you for your thoughtful comments and feedback! Here, we respond to specific questions and concerns the reviewer raises.
>
>
> ---------------------------------------
> **Question/Comment 1**: It would be nice to see LIV performance on the ALOHA dataset.
>
> **Response 1**: Thanks for this suggestion. We have now added an updated Figure 3 to include comparison between GVL (Zero-Shot) vs. LIV on our ALOHA-250 dataset. As shown, LIV’s qualitative behavior is similar to the one on OXE – the VOCs are close to a uniform distribution. In conclusion, GVL still produces task progress estimates that are higher quality than LIV on ALOHA-250, though the gap has become smaller compared to the results on OXE since the ALOHA dataset is sufficiently challenging for GVL too.
>
>
> ---------------------------------------
> **Question/Comment 2**: The use of VOC on the OXE evaluation is somewhat circular.
>
> **Response 2**: Thanks for this suggestion. To break up the circularity, we need to manually delegate a few datasets to be of “high-quality” to use VOC as a metric. To this end, we have selected RT-1 [1], Bridge [2, 3], DOBBE [4], and BC-Z [5] datasets because these datasets have successfully been used in prior works for large-scale imitation learning. The result is illustrated in Figure 10, Appendix F of our updated manuscript. As seen, on this subset of high-quality and diverse OXE datasets, GVL consistently generates highly positive VOCs; in contrast, LIV generates VOCs close to a uniform distribution, indicating that it struggles to accurately predict language-conditioned task progress on unseen robot videos. These results validate our experimental protocol of using VOCs as a metric for universal value functions. In addition, we have also improved the writing in Section 4.1 to make the distinction clearer.
>
>
> [1] Brohan, Anthony, et al. "Rt-1: Robotics transformer for real-world control at scale." arXiv preprint arXiv:2212.06817 (2022).
>
> [2] Ebert, Frederik, et al. "Bridge data: Boosting generalization of robotic skills with cross-domain datasets." arXiv preprint arXiv:2109.13396 (2021).
>
> [3] Walke, Homer Rich, et al. "Bridgedata v2: A dataset for robot learning at scale." Conference on Robot Learning. PMLR, 2023.
>
> [4] Shafiullah, Nur Muhammad Mahi, et al. "On bringing robots home." arXiv preprint arXiv:2311.16098 (2023).
>
> [5] Jang, Eric, et al. "Bc-z: Zero-shot task generalization with robotic imitation learning." Conference on Robot Learning. PMLR, 2022.
>
> ---------------------------------------
> **Question/Comment 3**: more analysis regarding GVL’s ability to accurately estimate values in videos where repetition is present.
>
> **Response 3**: Thanks for this suggestion! We have included two videos in the updated supplementary material in which repetition or failure is present. As shown, in both cases, when the robot gripper pulls back from the object of task interest, the GVL task progress estimates decrease, and when the gripper recovers and makes progress again, GVL estimates increase. These results demonstrate GVL’s ability to accurately estimate values in videos when repetition is present.
>
> ---------------------------------------
> **Question/Comment 4**: ALOHA-13 should be clarified.
>
> **Response 4**: Thank you for this suggestion. We have added a sentence in Section 4.2 clarifying. In short, ALOHA-13 is a subset of ALOHA-250 for which we have collected additional demonstration data. For example, instead of having only 2-3 demonstrations per task in ALOHA-250, we have 500+ Demos per task in ALOHA-13. This allows us to obtain statistically significant results on a task level for the in-context value learning experiments.
>
> We thank the reviewer again for their time and effort helping us improve our paper! Please let us know if we can provide additional clarifications to improve our score.

---

> ### Comment · Reviewer_RQdG · 2024-11-25
>
> Thank you to the authors for their detailed response! For the reasons discussed below, I will be further raising my score.
>
> **Question/Comment 1: It would be nice to see LIV performance on the ALOHA dataset.**
> Thank you for making the additional effort to gather these results. I also appreciate the honest discussion from the authors regarding the performance of GVL and LIV on ALOHA-250: "GVL and LIV exhibit similar qualitative trends,
> indicating that both methods now struggle with the complexity of this dataset."  That being said, GVL is able to augment its performance via in-context learning. In contrast (at least to my current understanding, however it would be helpful if the authors could verify), LIV cannot in-context learn due to its architecture: it computes values using cosine similarity in the embedding space rather than using a generative method. Due to the performance benefits of in-context learning, which only GVL can achieve, GVL seems to have a clear edge.
>
> **Questions/Comments 2-3: The use of VOC on the OXE evaluation is somewhat circular. More analysis regarding GVL’s ability to accurately estimate values in videos where repetition is present.**
> Thank you for addressing these concerns. I appreciate the isolation of specific "high-quality" datasets from OXE; doing so circumvents the issue with circularity and provides solid evidence of GVL's performance.
>
> That being said, it crossed my mind that perhaps GVL has simply learned to output a linear, sequential ordering (from initially shuffled data) rather than truly evaluate observations. I don't, however, believe that this is the case. Your two case-studies are helpful for intuitively visualizing what is going on, but my primary reason for believing that GVL works is GVL-SD. Since GVL is able to more effectively filter datapoints than the previous baseline (SuccessVQA), it seems clear to me that GVL can effectively assign values to "bad" trajectories (which would result in low VOCs). Thus, the success of GVL-SD seems to indicate GVL's ability to properly evaluate trajectories rather than simply output linear, sequential orderings.
>
> For future work, I think it would be good to gather a larger dataset of ground truth "bad" trajectories (i.e., ones that contain repetition or reverse progress) for a more systematic evaluation. Right now GVL's ability needs to be implied from the success of GVL-SD and a few case-studies. However I think this inference is reasonable and makes sense for the scope of this paper.
>
> **Question/Comment 4: ALOHA-13 should be clarified.**
> Thank you to the authors for clarifying this and for adding more description to the paper.

---

### Author Response · Authors · 2024-11-22
**Updated Paper and Experiments**

Dear all reviewers and AC,

We thank you for your time and effort in reviewing our work and providing detailed suggestions in improving our work! We have again updated our manuscript (all edits are highlighted in red color) and added several new evaluation results based on the anonymous reader’s feedback. In summary, our new results include:

1. New real-world experiments demonstrating how GVL can be used to generate effective co-training datasets from a raw, mixed-quality large dataset for real-world imitation learning (Section 4.3 and Appendix F)

2. New quantitative evaluation on a subset of OXE datasets validating our experimental protocol of using the introduced Value-Order Correlation (VOC) metric for evaluating GVL and baselines (Section 4.1 and Appendix F)

3. New qualitative results that demonstrates GVL’s ability to accurately estimate values in videos where repetition or failure is present (Supplementary Material and Appendix F)

Below, we have also responded to each individual reviewer’s questions, and we are eager to receive the reviewers’ reply! Please let us know if there are any lingering questions and we are available and happy to address any further inquiries you may have.

Best wishes,

Authors

---

### Meta-Review · Area_Chair_eEFM · 2024-12-19

**Metareview:**

The paper presents Generative Value Learning (GVL), a novel approach leveraging Vision-Language Models (VLMs) to predict task progress from visual trajectories. The reviewers' comments and the authors' responses provide a comprehensive view of the paper's strengths and areas for improvement.
The paper has several notable strengths. The proposed GVL method is conceptually interesting and supported by extensive experiments across diverse real-world tasks. The authors effectively addressed the reviewers' concerns, improving the paper's quality and clarity. While the method's simplicity was initially noted as a potential weakness, the authors demonstrated its effectiveness through comprehensive evaluations and comparisons. Overall, the paper meets the standards for acceptance, as it presents a novel and valuable contribution to the field of vision-language models and robotics, with the potential to impact downstream applications in policy learning and dataset management.

**Additional Comments On Reviewer Discussion:**

### Strengths Acknowledged by Reviewers:
- The proposed technique shows effectiveness in various experiments, such as outperforming baselines (LIV) in value prediction for text-based goals and complex videos
- The paper is well-motivated, with a clear explanation of the limitations of prior works and the importance of a "universal value estimator."
The idea of framing value estimation as a temporal ordering problem over shuffled videos and introducing the Value-Order Correlation (VOC) metric is novel and well-suited to the problem.
- The experiments are extensive, covering a wide range of real-world tasks and applications, including dataset filtering, success detection, and visuomotor policy learning.

### Weaknesses and Authors' Responses:
- Circularity in VOC Metric: The initial use of VOC on the OXE dataset was seen as circular. The authors addressed this by manually selecting "high-quality" datasets (RT-1, Bridge, DOBBE, and BC-Z) and showing that GVL consistently generates highly positive VOCs compared to LIV, which validates the experimental protocol.
- Analysis of GVL in Videos with Repetition: Reviewers requested more analysis in this regard. The authors included two videos in the supplementary material, demonstrating GVL's ability to accurately estimate values in the presence of repetition or failure.
- Limited Comparisons in Experiments: For visuomotor control tasks, the reviewer questioned the lack of comparison with LIV. The authors clarified that the purpose was to compare different dataset compositions generated by GVL rather than different visual representations. They used state-of-the-art algorithms (ACT and DP) with fixed vision encoders (Resnet-18) and compared variants based on training datasets.
- Real-World Results and Policy Learning: The real-world results' improvement over the baseline was considered small. The authors added an experiment showing that GVL's data quality estimation can discover better co-training datasets from raw datasets like DROID, enabling diffusion policies to match or outperform baselines.

---

### Decision · Program_Chairs · 2025-01-22

Accept (Spotlight)